# The acceptability and effect of a culturally-tailored dance intervention to promote physical activity in women of South Asian origin at risk of diabetes in the Netherlands— A mixed-methods feasibility study

Erik Beune[1]*, Mirthe Muilwijk[1], Judith G. M. Jelsma[2], Irene van Valkengoed[1], Annemarie M. Teitsma-Jansen[1], Bernadette Kumar[3], Esperanza Diaz[3,4], Jason M. R. Gill[5], Anne Karen Jenum[6], Latha Palaniappan[7], Hidde P. van der Ploeg[2], Aziz Sheikh[8], Emma Davidson[9], Karien Stronks[1]

1 Department of Public and Occupational Health, Amsterdam Public Health Research Institute, Amsterdam UMC, Location AMC, University of Amsterdam, Amsterdam, The Netherlands, 2 Department of Public and Occupational Health, Amsterdam Public Health Research Institute, Amsterdam UMC, Vrije Universiteit Amsterdam, Amsterdam, The Netherlands, 3 Unit for Migration and Health, Norwegian Public Health Institute, Oslo, Norway, 4 Department of Global Public Health and Primary Care, University of Bergen, Bergen, Norway, 5 Institute of Cardiovascular and Medical Sciences, University of Glasgow, Glasgow, United Kingdom, 6 Department of General Practice, Institute of Health and Society, General Practice Research Unit (AFE), University of Oslo, Oslo, Norway, 7 Department of Medicine, Stanford University School of Medicine, Stanford, California, United States of America, 8 Centre for Population Health Sciences, Usher Institute, University of Edinburgh, Scotland, Edinburgh, United Kingdom, 9 Centre for Clinical Brain Sciences, University of Edinburgh, Scotland, Edinburgh, United Kingdom

* e.j.beune@amsterdamumc.nl

## Abstract

### Objective

Populations of South Asian (SA) origin are at high risk of type 2 diabetes (T2D) and related complications. Analysis of T2D prevention interventions for these populations show that limited attention has been given to facilitating increased physical activity (PA) in a culturally appropriate manner. The aim of this feasibility study was to identify whether culturally tailored dance is acceptable to women of SA origin, and whether it may have an effect on PA and PA-related social cognitive determinants.

### Methods

A community-based culturally tailored dance intervention choreographed to Bollywood music was evaluated among 26 women of SA origin in the Netherlands for 10 weeks, 2 times per week. This feasibility study was conducted as a before-after, mixed-methods study, combining data from focus groups, individual interviews, questionnaires and accelerometers.

**Data Availability Statement:** The EuroDhyan study data are owned by the Amsterdam University

Medical Centers, location AMC in Amsterdam, The Netherlands. Data cannot be shared publicly because of privacy protection rules. Data are available on request for researchers who meet the criteria for access to confidential data and when the request fits within the goals of EuroDhyan. Requests can be sent to Dr. Henrike Galenkamp, Scientific Coordinator and Data Manager at the Amsterdam University Medical Centers, location AMC-Dept. of Public and Occupational Health, Room J2-225, Meibergdreef 9 1105 AZ Amsterdam, +3120 566 6450 at: h. galenkamp@amsterdamumc.nl. All requests will be processed in the same manner.

**Funding:** This work was funded by The Health Program 2014-2020 from the European Union, grant number 664609 HPPJ-2014, set up to improve the prevention of diabetes in South Asians. The funders had no role in study design, data collection, analysis, data interpretation or writing of the paper.

**Competing interests:** The authors have declared that no competing interests exist.

## Results

The majority of participants were in the age of 50–59 years and at moderate-to-high T2D risk. There was high attendance (73%), low drop out (12%) and high satisfaction scores for various program components. Key reasons for participation were the cultural appropriateness, in particular the combination of historically and emotionally embedded Indian music and dance, and the non-competitive nature of the intervention. On average, in each of the 19 one-hour sessions, participants spent 30.8 minutes in objectively assessed light intensity PA, 14.1 minutes in moderate intensity PA and 0.3 minutes in vigorous PA, and took 2,100 steps during a session. At follow-up, total moderate-to-vigorous PA increased by 7.8 minutes per day (95%CI: 3.1, 12.5) and daily steps increased by 784 (95%CI: 173, 1394), with a concomitant reduction in light PA of 12 minutes per day (95%CI: -21.9, -2.2). Positive shifts in some PA-behavioral social cognitive determinants and no negative side-effects were observed.

## Conclusions

Bollywood Dance appears to be a feasible, culturally acceptable and potentially effective approach to increase PA in SA women in the Netherlands. A pilot cluster RCT is needed to confirm these initial findings on effectiveness.

## Introduction

Over five million people of South Asian (SA) origin live in Europe (e.g. of Bangladeshi, Indian, Pakistani, Sri Lankan ancestry) [1]. Populations of SA origin are at increased risk of type 2 diabetes (T2D) and related complications compared to European-origin populations [2, 3]. Their risk of T2D is up to six times higher than populations of European origin [2, 3], and their proportion of deaths attributable to T2D is almost 50% higher than that in European-origin populations [4]. In addition, there are no indications that the risk will level off over time or across younger generations, with the increase of future T2D disease burden for this group being predicted to be even greater than projections for majority populations [5]. Previous studies have shown that intensive lifestyle interventions, such as diet and physical activity (PA), may be effective in producing consistent changes in glucose measures, weight and waist circumference which can assist in preventing the onset of T2D [6–8]. However, our analysis of available interventions for T2D prevention in populations of SA origin in Europe has found that there is limited experience of PA interventions specifically adapted for people of South Asian origin [9, 10], and within the studies in South Asians including a PA component, the PA element is often underused by participants (e.g. high drop-out rates) [9, 11]. WHO guidelines on physical activity and sedentary behavior recommend that all adults undertake 150–300 minutes of moderate-intensity or 75–150 minutes of vigorous-intensity aerobic physical activity per week or an equivalent combination of these [12]. PA is specifically one of the major recommendations for the prevention of T2D, however, low levels of PA have been reported among populations of SA origin [13]. SA women in particular are prone to low PA levels due to perceived barriers, acculturative stressors and cultural priorities regarding health [13]. These factors highlight the need for developing and testing PA interventions that are acceptable to participants and thereby may increase T2D prevention effectiveness [10].

Psychological, social and cultural factors (e.g. self-efficacy, social support, perceived social norms) play an important role in PA initiation and maintenance and should be taken into account when designing PA behavior change interventions [14].

Qualitative studies report that culturally tailored dance is perceived as a motivating form of PA by some populations of SA origin [15, 16]. Populations of SA origin have long enjoyed a rich culture involving a high-energy, popular dance style called Bollywood, inspired by India's film industry [17]. Two previous PA intervention studies based on Bollywood dance for women of SA origin in Canada and the United States of America (USA) showed promising results, i.e. excellent participation [17, 18], and satisfaction rates [18], and small, but significant improvement in HbA1C [17]. Furthermore, those women with high participation rates in the dance classes showed reductions in weight [17].

Bollywood dance might be a culturally acceptable type of PA to be used in T2D prevention in populations of SA origin in Europe, although before implementing Bollywood dancing as a health promoting PA intervention investigation of its acceptability is needed. The main aim of this mixed-methods feasibility study was to identify whether and how Bollywood dancing as a health promoting intervention (henceforth "Bollywood Dance Fitness") may help to increase PA among women of SA origin residing in the Netherlands. In particular, we explored the following research questions: 1) whether Bollywood Dance Fitness was an acceptable form of PA to motivate women to participate in a PA promoting program, and whether the program succeeded to keep participants attending; 2) what various actors involved (i.e. participants, instructor, community leader) experienced and perceived as successful and less successful elements; 3) to what PA intensity levels participants were exposed during Bollywood Dance Fitness; and 4) what the effect was on participant's PA-level, PA-compensation behavior and PA-related social cognitive determinants.

## Materials and methods

### Study design

Bollywood Dance Fitness was evaluated using a single-arm before-after, mixed-methods study design, combining data from questionnaires, attendance sheets, accelerometers, focus groups and individual interviews, as outlined in Table 1 for each research question (RQ). This mixed-methods design is aimed to validate findings across different data sources. This study was part of larger project (i.e. EuroDHYAN) [8, 19], which aimed to facilitate the development of novel and targeted health promotion strategies to effectively reduce the risk of T2D in SA origin populations in Europe.

### Study population

The study took place in Amsterdam, the Netherlands. Participants were members of the Surinamese community of SA origin, which refers to people of South Asian ancestral origin and their offspring who migrated to the Netherlands via Suriname. The Surinamese community of SA origin are the descendants of the laborers from North India—Uttar Pradesh, Uttaranchal, and West Bihar–who were indentured between 1873 and 1917 [20]. Surinamese SA origin form a distinct ethnic group within the Netherlands. They number about 150,000 and approximately 65% are Hindus, 25% Muslims and 5% Christians. The majority has a low-middle socioeconomic level [21].

Surinamese women of SA origin (born and/or one parent born in Suriname, self-identified SA origin) aged between 30–65 years and without known diabetes were recruited via a local community center for Surinamese of SA origin, through flyers at local pharmacies in South East Amsterdam, and subsequent snowball sampling. To evaluate whether Bollywood Dance

**Table 1. Data collection method for each research question (RQ).**

| | Quantitative methods | | | | Qualitative methods | | |
|---|---|---|---|---|---|---|---|
| | **Baseline and post program questionnaires participants (n = 26)** | **Participant attendance sheets from each participant (n = 26) for each session (n = 19)** | **Accelerometer from each participant (n = 26) from each session (n = 19) + from dance instructor from 3 sessions** | **Accelerometer from each participant 7 days at baseline (one week prior to the start of the intervention) and at follow-up (last week from the intervention (n = 19 valid))** | **Post-program semi-structured interview (n = 1, 45min) with key leader of community center (n = 1)** | **Mid- and post-program semi-structured interviews (n = 2, 45 min) with instructor (n = 1)** | **Post-program Focus Group Discussions (n = 2, 1½ hr) with participants (n = 18)** |
| RQ1 Acceptable form of PA and keep women attending | X | X | | | X | X | X |
| RQ2 Experiences with the program | X | | | | X | X | X |
| RQ3 To what PA level exposed | | | X | | | | |
| RQ4 Effect of the program | X | | | X | | | |

Fitness may be feasible and acceptable to be used in T2D prevention in at-risk populations, this study specifically included persons without known diabetes. Persons with any serious medical condition which would prevent long-term participation or which would contraindicate PA, current atrial fibrillation, active treatment for cancer, cognitive inability as judged by the interviewer, living in a nursing home or on the waiting list for a nursing home, having a physical disability and pregnant women were excluded. All potential participants were invited to attend a baseline measurement.

### Intervention

To recognize shared beliefs, values, and practices of women as part of the culture of the Surinamese community of SA origin, Bollywood Dance Fitness was developed in collaboration with the community. A core group of people from the Surinamese community of SA origin co-created the Bollywood Dance Fitness program, informed by input from: a) Surinamese SA origin key leaders and potential end-users; b) a female SA origin Bollywood dance instructor (DD); and, c) the PI (LP) from a Bollywood dance initiative in the USA. Given that the USA Bollywood dance initiative was eight weeks in women with T2DM [17], LP advised a longer intervention in at-risk populations, to examine adherence and satisfaction, and to increase generalizability.

The Bollywood Dance Fitness consisted of 20 planned dance classes of one hour each, twice weekly (Mondays and Thursday evenings) for 10 weeks. The classes comprised: warming up (5 min), aerobic dance (30 min) choreography and music derived from popular 'Bollywood movies', strength training exercises (15 min) and cooling down (10 min) and were led by a female dance instructor (DD). This was supplemented by three 30-min goal setting education

sessions (in weeks 3, 5, 7) led by EB. Implementation was supported by a written protocol for the dance instructor combined with two 120 minutes face-to-face training sessions provided by EB, with the second one including feedback, based on structured observations from some of the dance classes, informed by a topic-guide including context (barriers, facilitators), participants (performance, intensity, pleasure, confidence, social interaction) and teaching (attitudes, performance, learning strategies, adherence to protocol). The venue was located in the close neighbourhood of most participants.

Resnicow has described cultural sensitivity by two dimensions, i.e. 'deep structures' and 'surface structures' [22]. The intervention comprised deep structure (e.g. Indian music, cultural dance) and surface structure adaptations (e.g. instructor of SA origin) [22]. The intervention addressed relevant social cognitive determinants of beginning and continuing a PA program, i.e. outcome expectations, self-efficacy, capability and skills, perceived norms and behavior of others and social support [14]. Alongside these determinants, the main approaches employed during the program were active learning, reinforcement, modeling, direct experience, guided practice, mobilizing social support and goal setting [23].

A community key leader (director of Surinamese SA origin community center) provided support for making available the venue (community center meeting room) and a person for opening/closing/assistance for organisational-related issues and free tea/coffee. Further, a text message group (through mobile phone) for the participants and the instructor was created, to facilitate interaction and support throughout the program (e.g. input/share favorite music for dance sessions, experiences from the dance sessions (photos, videos) and/or any questions).

## Data collection

Data were collected between September 2017 and January 2018.

**Qualitative data.** Two post-intervention focus group discussions (FGD) (n = 10, n = 8, respectively) conducted by EB (moderator) and AT (co-moderator) with participants, two interviews (mid-term and post-intervention) conducted by EB with the Bollywood Dance Fitness instructor (DD), and one post-intervention interview conducted by EB with the community leader (AB), were used for data collection on acceptability, feasibility and experiences with the program. All interviews were held at the community center, in Dutch. All participants spoke fluently Dutch, as in Suriname Dutch is still the official and prevailing language of government, business, media and education. The interviews were informed by semi-structured topic-guides (S1 Table). All interviews were audio-recorded and transcribed verbatim without the use of audio transcription software.

**Quantitative data.** *Structured questionnaires and attendance sheets.* Face-to-face structured questionnaires were collected at baseline, administered on paper (one week before intervention) and follow-up (1–3 weeks after the intervention). We measured: 1) socio-demographics, including age, perceived general health and physical disability, FINDRISC-score [24], ethnicity, duration of stay in the Netherlands, educational level, religion, marital status, household situation, employment status (baseline only); and 2) with questionnaires derived from the DHIAAN study [20], PA-behavioral social cognitive determinants including attitude, social support, social norms, self-efficacy, the stage of change towards PA and capability and skills towards goal setting/dancing skills, with all items measured on a 5-point Likert scale (from completely disagreeing to completely agreeing with different statements) (baseline/follow-up); and 3) experiences with the intervention, including satisfaction with components (5-point scale), cultural appropriateness (5-point scale), negative/positive (side) effects (5-point scale), reasons for non-attendance, intention to continue (yes/no), related costs (follow-up only). A satisfaction score for each of the components of Bollywood Dance Fitness

program was calculated and a frequency count was used to determine cultural appropriateness, perceived benefits or negative (side) effects and intention for continuation with Bollywood Dance Fitness and appreciation of characteristics of the instructor. Participant attendance sheets were collected by the dance instructor for each session during 10 weeks.

*Physical activity measures.* Physical activity was measured using accelerometers (Actigraph: GT3X+; Pensacola, Florida, USA) for seven days at baseline (one week prior to the start of the intervention) and during the follow-up measurement (in week 10 of the intervention). Participants were instructed to wear the accelerometer on an elastic belt positioned over the right hip during waking hours and only remove the device for water-based activities (e.g. swimming, showering). Activity diaries were kept to report date, time out of bed, start time and end time of wearing the accelerometer, time to bed, reasons for non-wear/removal, and time spend on swimming activities. Raw data were analyzed in 60-second epochs and cleaned using the Actilife software (version 6.8.2). The choice of 60-second epochs was based on the advice to use the epoch cut off with which the cut offs were designed [25]. Troiano's NHANES wear time validation definitions were used [26]. A day was considered valid if the accelerometer was worn for 10 hours or more, whereby bouts of $\geq$ 60 minutes of zeros (allowing two interruptions in the zeroes between 1–100 counts/min) were considered as non-wear time. Individual accelerometer files were also manually checked against the activity diaries and data outside the wearing timeframe indicated in the diary were removed. Participants who had at least four days of valid data were included in the analyses [25, 27]. During each of the one-hour Bollywood Dance Fitness sessions, all participants also wore an accelerometer to measure PA. The dance instructor wore an accelerometer as well, but in session 17, 18 and 19 (week 9 and 10) only. This made it possible to identify the potential intensity-level if participants would achieve ideal performance (like the instructor). Raw data of the Bollywood Dance Fitness-session were also analyzed in 60-second epochs in the Actilife software. For all collected accelerometer data, mean counts per minute (CPM) were converted to minutes in sedentary, light, moderate, and vigorous intensity PA based on the Troiano NHANES cut-off points [26]; sedentary <100 CPM; light intensity between 100–2019 CPM; moderate intensity between 2020–5998 CPM; vigorous intensity 5999 and above CPM. Moderate and vigorous activity were summed and presented as moderate and vigorous physical activity (MVPA). Light, moderate, and vigorous PA were summed and presented as total PA. Sedentary, light and total PA were also presented as percentage of total wear time, as longer wear time leads to more time to accumulate PA (only for the 7-day data). Furthermore, total number of steps a day (or per session) and the percentage of participants meeting the PA guidelines [12], of 150 minutes MVPA per week (= 21.4 min/day), were calculated.

### Data analysis

**Qualitative analysis.**   Transcripts of the focus group discussions and individual interviews were coded through qualitative thematic analysis (deductive and inductive coding) by EB [28]. To answer RQ 1 and RQ2 (Table 1), transcripts were open coded, codes were sorted into themes, and then interpreted by EB and AT whereby possible themes were collapsed. MaxQDA Analytics Pro (VERBI Software, 2016) was used for data analyses to process, order and compare the codes.

**Statistical analysis.**   To determine the exposure to PA during the program (RQ3), PA in the different intensity categories were modelled for each session and with generalized estimating equations (GEE) analyzed with an exchangeable structure. The GEE evaluated if an increase in intensity over time existed and if this increase was dependent on amount of

sessions attended. Furthermore, the PA data are presented for each participant averaged over all attended sessions and for the dance instructor averaged over the three measured sessions.

To investigate the effect of the program for PA level (RQ4) paired-samples t-test were used to evaluate differences in the accelerometer data between baseline and follow-up for PA level, both analyses for total PA with and without the dance sessions included at follow-up are presented. Secondly, the proportion of participants who achieved the level of PA recommended in the Dutch PA guideline (150 min/week of MVPA is operationalized to 21.4 min/day) was calculated [12], whereby the difference between baseline and follow-up was evaluated with the McNemar Test.

To investigate the effect of the program for PA determinants (RQ4) the proportion agreeing, on an ordinal scale from completely disagreeing to completely agreeing, with different statements at baseline and follow-up was calculated, whereby the difference between baseline and follow-up was evaluated with the Wilcoxon signed-rank test for paired measurements.

R studio version 0.99.903 [29] and for accelerometer data SPSS version 22.0 [30] were used for statistical analyses.

### Ethical considerations

The Medical Ethical Committee of the Amsterdam UMC, location AMC reviewed the study protocol and stated that no further assessment and approval from any officially accredited Medical Ethical Research Committee was needed (reference number 09171260). In line with the Amsterdam UMC, location AMC code for good conduct of medical research provisions were made to assure the respondents anonymity in collection, storage, analysis and presentation of the data. Written informed consent was obtained from all participants for the digital recording of the interviews, the questionnaire and physical data. They were assured anonymity in the presentation and publication of the data.

## Results

We describe results on recruitment and participant characteristics and whether the program was delivered as planned. Next, we describe participants' views on attractiveness and acceptability followed by program attendance, retention and experiences (RQ1 and RQ2). Finally, we present results on PA intensity (RQ3) and effectiveness of the program on PA and PA determinants (RQ4). The qualitative findings are illustrated by literal citations of participants' answers in interviews and are used to inform observed findings in quantitative data.

### Recruitment and participants characteristics

Fig 1 presents the flow diagram of the participants. Thirty-seven women of SA origin, aged between 30–65 were recruited (15% from local community center, 12% from local pharmacies, 73% snowball sampling). After screening for eligibility, five women were excluded due to T2D diagnoses, five women were unable to attend the program at the available time slot, and one woman was impossible to contact. Eventually, 26 SA women (70%) were available to participate.

Table 2 describes the characteristics of the 26 participants at baseline. All SA origin women were born in Suriname. The majority were in the age of 50–59 years, were Hindus, at moderate-to-high T2D risk based on FINDRISC-score, had fair-to-good perceived health and had resided on average 31.8 years in the Netherlands. Furthermore, half of the women (54%) had a paid job whereas the educational level was fairly equally distributed among the women.

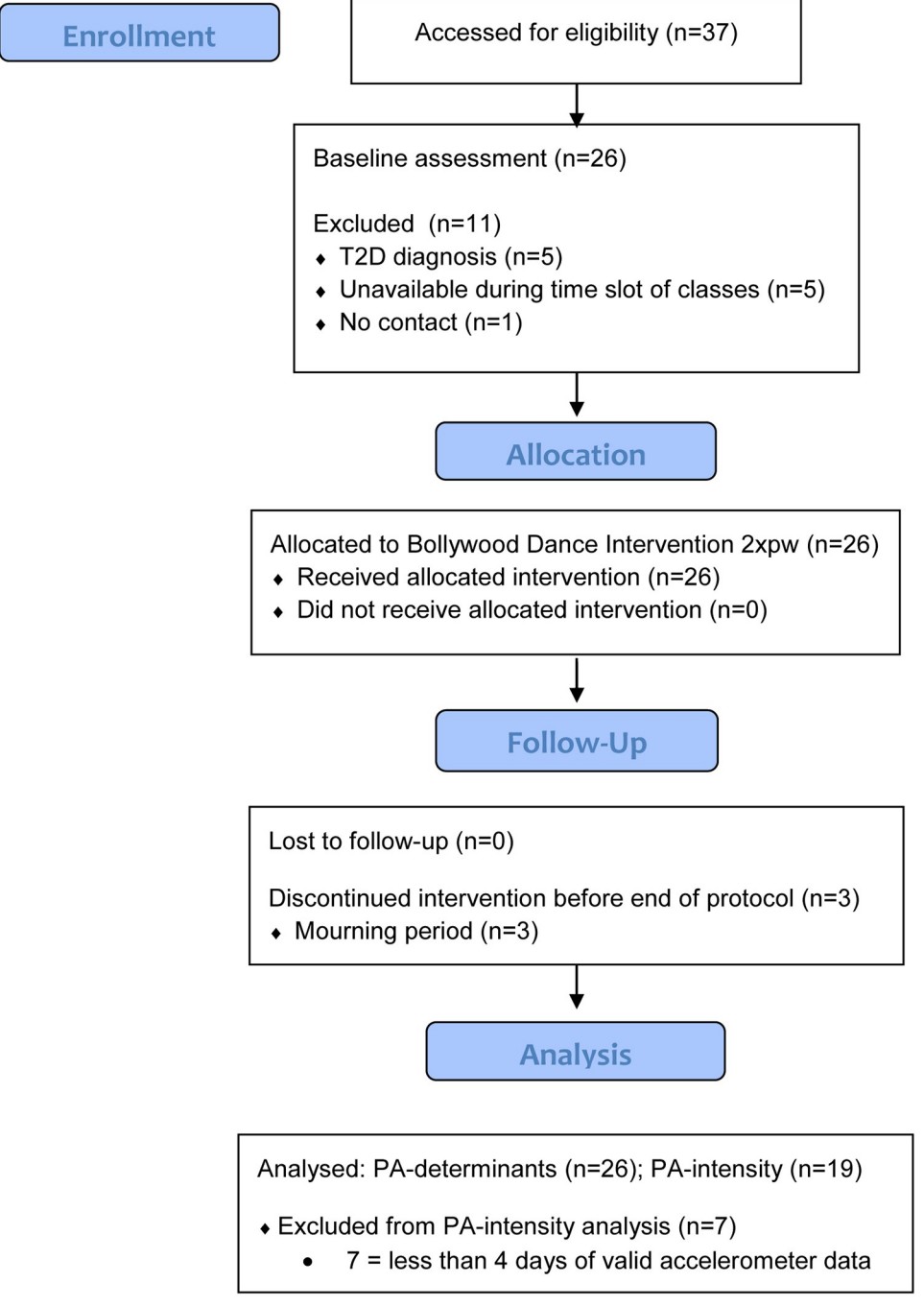

**Fig 1. Flow diagram of the participants.**

## Program delivery

The intervention ran from October 12th to December 18th 2017. In total, 19 out of the 20 intended sessions (including dance/aerobics and resistance exercise) were delivered. One session had to be cancelled due to extreme snowfall. All three intended goal setting lessons were delivered.

**Table 2. Baseline characteristics women of South Asian origin (n = 26).**

| | |
|---|---|
| **Age: (years) n (%)** | |
| 30–39 | 4 (15) |
| 40–49 | 3 (12) |
| 50–59 | 15 (58) |
| 60–65 | 4 (15) |
| **Duration of stay in NL (years): mean (SD)** | 31.8 (11.1) |
| **Education level: n (%)** | |
| Low | 8 (31) |
| Middle | 10 (38) |
| High | 8 (31) |
| **Employment status: n (%)** | |
| Paid work | 14 (54) |
| Unpaid work / housewife | 7 (27) |
| Unemployed/disabled | 4 (15) |
| Retired | 1 (4) |
| **Marital status: n (%)** | |
| Married | 6 (23) |
| Not married | 10 (38) |
| Divorced | 8 (31) |
| Widow/other | 2 (8) |
| **Religion: n (%)** | |
| Hindu | 18 (69) |
| Islam | 5 (19) |
| Christian | 1 (4) |
| Other | 2 (8) |
| **Family History of diabetes: n (%)** | |
| Yes | 22 (85) |
| **FINDRISC[1] score: n (%)** | |
| < 7 (low) | 1 (4) |
| 7–11 (slightly elevated) | 7 (27) |
| 12–14 (moderate) | 8 (31) |
| 15–20 (high) | 8 (31) |
| > 20 (very high) | 2 (8) |
| **Body Mass Index[2] (kg/m2)** | |
| 18–22,9 (normal weight) | 6 (23) |
| 23–24,9 (overweight) | 1 (4) |
| ≥25 (obese) | 19 (73) |
| **Waist circumference (cm) n (%)** | |
| < 80 (normal risk) | 0 (0) |
| 80–88 (increased risk) | 7 (27) |
| > 88 (substantially increased risk) | 19 (73) |
| **Perceived health: n (%)** | |
| Fair | 11 (42) |
| Good | 12 (46) |
| Very good/excellent | 3 (12) |
| **Physical complaints: n (%)** | |
| No | 7 (27) |
| Yes, moderate | 12 (46) |

(*Continued*)

**Table 2.** (Continued)

| | |
|---|---|
| Yes, severe | 1 (4) |
| missing | 6 (23) |
| **Stage of change for PA: n (%)** | |
| Pre-contemplation | 6 (23) |
| Contemplation | 5 (19) |
| Preparation | 4 (16) |
| Action | 0 (0) |
| Maintenance | 11 (42) |

[1]Finnish Diabetes Risk Score (FINDRISC) [24] adapted by
[2]Ethnic-Specific Criteria for Classification of BMI [37].

## Attractiveness and acceptability

We found that for women of SA origin the direct fit in their lifestyle and culture, and its non-competitive nature were key characteristics of the program. These key characteristics gave the women a feeling of safety and security and had tempted them to this particular PA initiative.

> *"It was a group of SA women, about the same age, you feel comfortable in that group, there is no competition. If you go to a gym, then I'm watching how everyone is muscled and showing off, and how everyone, then you think: 'what's this!?' And that's what I do not want to participate in!"*
>
> *(FGD2; ID23; 57yr)*

Women emphasized that the perspective of Indian songs and dances had attracted them, due to their emotional and historical connectedness with it. Also the instructor emphasized the Bollywood music and dance was an integral part of lifestyle, history and culture among women of SA origin.

> *"This [the songs] is of course what they grew up with, they understand all the lyrics, it does something with them, I think."*
>
> *(II_Instructor-post program)*

Women explained and underscored that the fact the sessions were for women only made the program more attractive and socio-culturally acceptable for women of SA origin to take part in Bollywood Dance Fitness.

> *"I think when it will come out that I dance with men in a group, that this will be interpreted wrongly by others"*
>
> *(FGD2; ID4; 30yr)*

In addition, participants indicated that the familiarity with the community center of Surinamese South Asians contributed to attractiveness and acceptability of the PA program.

The questionnaire data indicated that the majority of the participants perceived the intervention as culturally appropriate; i.e. 88% of the women felt that the dance lessons matched good-to-very good with their cultural background and 70% felt that cultural adaptation of the intervention was important-to-somewhat important.

## Attendance, retention

Attendance logs showed that per session on average 73% of the women participated. This also included one session where only eight women were present, because of the 'Diwali' festival (traditional Hindu celebration), but the group had decided that lessons should continue for those that could attend. Both questionnaires and attendance logs indicated family-related circumstances and illness as main reasons for non-attendance. Drop out was low throughout the whole program: only three participants (12%) dropped out, all due to death of a close family member and the following mourning period.

The interviews and focus groups showed that participants mainly had positive experiences during the dance sessions, which may explain the high attendance and retention rates. In particular, the high 'fun factor' kept participants motivated to continue, together with increased dancing capabilities, self-esteem and the enthusiastic, culturally knowledgeable and dedicated dance instructor.

> *"In the beginning when we had conversations then everyone said, those Bollywood movies, I can't, I can't. And because she [the dance instructor] has built it up, in the end we could see that everyone could do it!"*
>
> *(FGD1; ID6; 52yr)*

Participants felt that the group-based nature in particular contributed to a feeling of belonging and solidarity for other group members and thereby created a social stimulant keeping them motivated to attend.

> *"If I had to go to the gym in this weather, in these times, I would choose not to do it. And here you do it with each other you know, that stimulates in one way or another"*
>
> *(FGD1; ID12; 51yr)*

The dance instructor explicitly mentioned that–*in line with the co-creative/bottom-up approach*—input from participants was appreciated and stimulated throughout the program (e.g. choice of music, choreography, dance moves) and that this might have been an important reason for the high retention rates. She clarified that these feedback loops made it possible to adapt the sessions to the taste and level of the group, stimulated active involvement and a feeling of being taken seriously.

> *"The women could indicate what they found difficult or liked, so they had input all the time, it was upon their request, so not all came from my side"*
>
> *(II_Instructor- post-program)*

Another reason for continued participation was the absence of any financial contribution. This was very much appreciated and the women stressed this as a factor that facilitates easy access and low drop out, in particular by people living on a tight budget.

> *"I think that otherwise [in case of financial contribution] many people will drop out and cannot contribute"*
>
> (FGD1; ID3; 57yr)

## Positive experiences with program components

In particular, the familiarity with the Indian music was mentioned by the women to be very helpful in supporting their PA behaviour. Women felt emotionally attached to Bollywood music and therefore perceived the music as a way of expressing happiness and joy, which directly stimulated them to move. They also felt that this music enhanced their memory and recognition of certain dance steps, facilitated automatic motor skills and the learning of new skills and helped them to push their limits.

*"When the music went on and the first pass begins, you just saw, shiny happy faces, immediately. Yes, we were allowed to dance, that was so nice. We were allowed to shake those buttocks!"*

*(FGD2; ID16; 56yr)*

The group-based nature facilitated copying dance moves from other women and a kind of collective memory, which made the dancing easier.

*"I danced with a group, and therefore I knew exactly which moves followed"*

*(FGD2; ID1; 63yr)*

Participants indicated that the Bollywood Dance Fitness sessions contributed to their well-being. Physical health in particular, i.e. feeling of having more energy, stamina, muscle power and easier participation in daily activities (i.e. climbing stairs, walking). Mental health effects were also mentioned, e.g. stress reduction and growth of self-esteem, while some women mentioned positive effects on their social life, such as extension of social network. None of the women mentioned any negative effects on their physical or mental status.

*Interviewer*: *What did it bring you?* *Participant*: *"My network of friends has expanded"*

*(FGD2; ID16; 56yr))*

Questionnaire data corroborated the positive contribution of dance sessions to participants' health and wellbeing as, i.e. 77% of the participants reported positive effects, such as increased strength/stamina and better relaxation/sleep.

Additionally, the questionnaire data showed high satisfaction with components of the Bollywood Dance Fitness (Table 3) and that over half (53%) of the women intended to continue with Bollywood Dance Fitness individually at home. All participants had the intention to continue with Bollywood Dance Fitness group sessions in the future. The most important prerequisites to proceed were continued availability in the close neighborhood and the convenient planning (evening/not during weekends). Although this pilot was free of charge, most participants (77%) were prepared to pay a limited amount (max. €2-3/session) for future dance sessions.

## Negative experiences with program components

Qualitative data showed that twice a week a dance session meeting was a bit challenging and caused some stress to get this organized in the personal life of some of the women. Secondly, when all 26 women were present, the venue was perceived as somewhat small. Thirdly, women would have liked to receive direct feedback about their PA level from the accelerometers throughout the session.

**Table 3. Satisfaction score[1] for Bollywood Dance Fitness components.**

| Program components | Mean (SD) | Range |
|---|---|---|
| Music | 4.85 (0.46) | 3–5 |
| Dance | 4.85 (0.46) | 3–5 |
| Resistance exercise | 4.77 (0.43) | 4–5 |
| Goal setting educational sessions | 4.70 (0.59) | 3–5 |
| Instructor | 4.85 (0.61) | 2–5 |
| Time of appointment | 4.58 (0.90) | 2–5 |
| Frequency (twice a week) | 4.85 (0.64) | 3–5 |
| Duration (1 hour) | 4.50 (0.99) | 2–5 |
| Intensity | 4.73 (0.53) | 3–5 |
| Venue (Community center) | 4.31 (1.01) | 2–5 |
| Other participants | 4.85 (0.46) | 3–5 |

[1]Note: scale 1 (minimum)– 5 (maximum).

Questionnaire data showed that two participants reported uncomfortable (side) effects (e.g. joint pain) of participation in the Bollywood Dance Fitness sessions.

### Program intensity and effectiveness

**PA intensity during the program.** On average, during the 19 one-hour sessions, participants spent 30.8 minutes in light intensity PA, 14.1 minutes in moderate intensity PA, 0.3 minutes in vigorous intensity PA, and the remaining 14.8 minutes were spent sedentary. Participants took on average 2100 steps during a session. Time spent in moderate to vigorous PA (MVPA) ($\beta = 0.198$; 95% CI: 0.072, 0.325) and step count ($\beta = 19.1$; 95% CI: 11.6, 26.4) increased gradually across the Bollywood Dance sessions. Furthermore, women who attended more lessons on average had slightly higher levels of MVPA during a session ($\beta = 0.045$; 95% CI: 0.009, 0.081).

There was a large variation between participants in the amount of moderate intensity PA during a session (Fig 2), which ranged from 0.8 to 25.2 minutes per session between participants. The dance instructor spent on average 25.3 minutes in light activity, 24.7 minutes in moderate activity and 9 minutes in vigorous activity during a session (based on sessions 17, 18 and 19), taking on average 3200 steps. When comparing the intensity data from the participants to that of the dance instructor, participants showed a significantly different distribution (one sample t-test) of PA over the intensity categories: i.e. light (mean difference: 4.2; 95%CI: 1.7, 6.7); moderate (mean difference: -7.8; 95% CI:-10.9, -4.8); vigorous (mean difference: -8.7; 95% CI: -8.9, -8.4); and also for steps per dance session (mean difference: -931; 95%CI: -1123.7, -738.4).

**Changes in PA and PA-compensation behavior after the program.** Accelerometer data (Table 4) showed that among participants MVPA increased at 10 weeks follow-up by 7.8 minutes per day (95% CI: 3.1, 12.5) and with an increase of 784 steps per day (95% CI: 173, 1394). This was matched by a decrease in light activity intensity PA by 12 minutes a day (95% CI: -21.9, -2.2). At baseline, one (5.3%) woman met the PA guidelines of at least 150 minutes of moderate intensity per week [31]; at follow-up, five women (26.3%) were physically active above the recommended level. Correction for percentage wear time showed similar results.

Separating the effect of participation in the Bollywood session from the follow-up results (deleting data from the Bollywood sessions) showed that the changes in MVPA and daily steps attenuated and were no longer statistically significant (Table 4). This suggests that

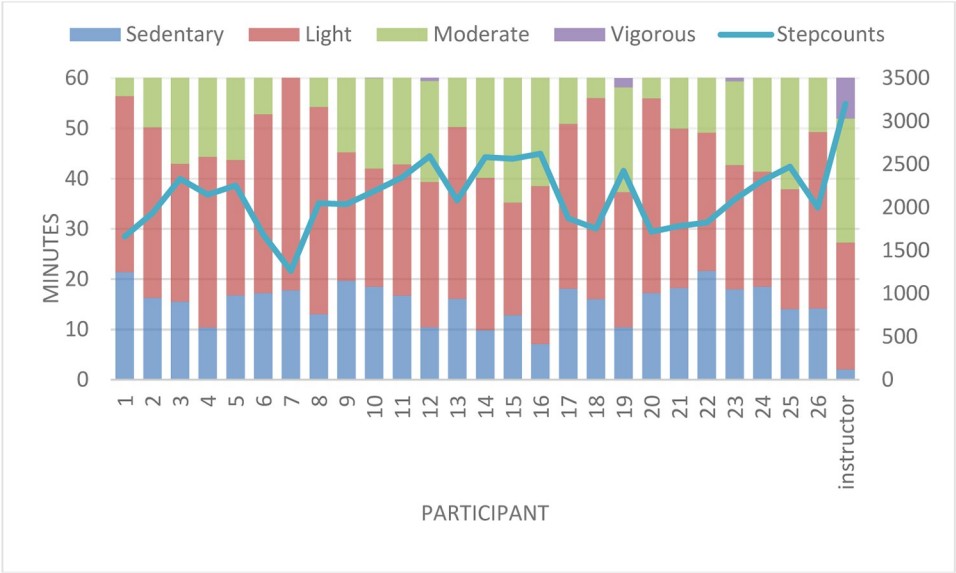

**Fig 2. Steps and averaged sedentary, light, moderate and vigorous intensity for each participant: All attended sessions.**

participation in the dance session was the primary reason for the improvements in MVPA and daily steps, but also that participants didn't compensate for participation in the dance session by being less active for the remainder of the week.

**Changes in PA-related determinants.** Self-reported questionnaire data (Table 5) showed at follow-up meaningful changes throughout the group in some PA-behavioral social cognitive determinants, in particular a positive shift in perceived PA skills (self-efficacy), stamina necessary to take part in Bollywood Dance Fitness (capability) and perceived social norms for PA (injunctive norm). In addition, positive changes in self-efficacy, skills, social norm and social support to PA were observed.

## Discussion

This study clearly demonstrated that a culturally tailored dance intervention choreographed to Bollywood music is acceptable for a subset of women of SA origin to be used as a PA intervention, with high satisfaction, attendance and retention levels to the 10 week PA program and corroborated results from previously conducted studies in Canada [18] (6 weeks), and the USA [17] (8 weeks). The direct fit of the PA program in the culture and the lifestyle of the SA women as well as its non-competitive nature were perceived as key elements for participation and continuation. The study also highlights that the Bollywood Dance Fitness program exposed participants on average to light-to-moderate PA intensity levels. After 10 weeks of Bollywood Dance Fitness, significant positive shifts were observed in daily minutes of MVPA (mainly through replacing light PA), in daily steps and in meaningful changes of some PA-behavioral social cognitive determinants. No unintended negative compensation behavior on PA, such as being less active outside of the dance classes, were observed.

Participation and retention in PA initiatives for women of SA origin has been considered a huge challenge, due to structural and socio-cultural barriers to PA [13]. Active user involvement of the target population (prior and during the PA intervention) was an essential element

**Table 4. The effect of Bollywood Dance Fitness on physical activity.**

| Variable | Baseline mean (± SD) | Effect with session | | | | Effect without session | | | |
|---|---|---|---|---|---|---|---|---|---|
| | | Follow-up mean (± SD) | Mean difference | 95% CI | p-value | Follow-up mean (± SD) | Mean difference | 95% CI | p-value |
| Physical activity, n | 19 | 19 | | | | 19 | | | |
| Activity counts (kilocounts/min) | **233.8 (62.8)** | **264.5 (58.6)** | **30.7** | **[3.0; 58.5]** | **0.032** | 243.1 (56.3) | 9.3 | [-15.8; 34.3] | 0.446 |
| Steps (steps/day) | **5821 (1456)** | **6605 (1545)** | **784** | **[173; 1394]** | **0.015** | 6036 (1425) | 214 | [-359; 788] | 0.443 |
| Weartime (min/day) | 790.2 (81.3) | 780.2 (63.0) | -10.0 | [-29.2; 9.1] | 0.286 | **764.4 (63.1)** | **-25.8** | **[-46.1; -5.6]** | **0.015** |
| Sedentary (min/day) | 486.9 (67.5) | 481.1 (49.1) | -5.8 | [-24.9; 13.3] | 0.532 | 477.4 (49.3) | -9.5 | [-28.9; 10.0] | 0.321 |
| Light (min/day) | **290.1 (53.4)** | **278.1 (48.5)** | **-12.0** | **[-21.9; -2.2]** | **0.019** | **270.1 (48.8)** | **-20.0** | **[-29.8; -10.2]** | **<0.001** |
| Light (% weartime) | 36.7 (5.5) | 35.6 (4.8) | -1.1 | [-2.3; 0.1] | 0.071 | **35.3 (4.8)** | **-1.4** | **[-2.6; -0.2]** | **0.023** |
| Moderate (min/day) | **13.2 (9.0)** | **20.9 (10.7)** | **7.7** | **[3.1; 12.3]** | **0.003** | 16.9 (10.3) | 3.7 | [-0.5; 7.9] | 0.084 |
| Moderate (% weartime) | **1.7 (1.2)** | **2.8 (1.5)** | **1.1** | **[0.5; 1.7]** | **0.002** | **2.3 (1.4)** | **0.6** | **[0.02; 1.1]** | **0.042** |
| Vigorous (min/day) | 0.0 (0.2) | 0.1 (0.3) | 0.1 | [-0.1; 0.3] | 0.230 | 0.0 (0.1) | 0.0 | [-0.1; 0.1] | 0.872 |
| Vigorous (% weartime) | 0.0 (0.0) | 0.0 (0.0) | 0.0 | [-0.01; 0.03] | 0.170 | 0.0 (0.0) | 0.0 | [-0.01; 0.01] | 0.993 |
| MVPA (min/day) | **13.3 (9.1)** | **21.1 (10.7)** | **7.8** | **[3.1; 12.5]** | **0.003** | 16.9 (10.3) | 3.7 | [-0.6; 7.9] | 0.086 |
| MVPA (% weartime) | **1.7 (1.2)** | **2.8 (1.5)** | **1.1** | **[0.5; 1.7]** | **0.001** | **2.3 (1.4)** | **0.6** | **[0.02; 1.1]** | **0.043** |
| Total physical activity (min/day)[1] | 303.4 (55.1) | 299.1 (48.3) | -4.2 | [-15.1; 6.6] | 0.423 | **287.0 (47.7)** | **-16.4** | **[-27.1; -5.6]** | **0.005** |
| Total physical activity (% weartime) | 38.4 (5.6) | 38.3 (4.8) | 0.0 | [-1.5; 1.4] | 0.983 | 37.5 (4.8) | -0.8 | [-2.2; 0.6] | 0.238 |
| n above the guideline, n (%) | 1 (5.3%) | 5 (26.3%) | | | 0.125 | 2 (10.1%) | | | 0.500 |

Notes: Bold numbers represent a statistically significant effect of p<0.05. Abbreviations: MVPA: moderate to vigorous physical activity; Effect with/without session: dance sessions in/excluded at follow-up (final week of the intervention).

[1]Total physical activity includes Light and MVPA.

of the current success and should be considered when such PA initiatives are to be applied elsewhere [9]. Furthermore, it is promising that participants perceived more acceptance by important others and felt more confident and capable to take part in PA initiatives after the Bollywood Dance Fitness, especially since injunctive norm and self-efficacy have been shown to be important determinants for PA participation [32]. As outlined in the methods, our study was based on behavioral social cognitive determinants identified by evidence synthesis, and Delphi expert consensus, to be important for the initiation and maintenance of PA [14, 23]. We measured changes in these PA-behavioral social cognitive determinants and our findings support some specific models of behavior change but are not consistent with others. For example, our results are in line with and underscore findings from the literature demonstrating that Ajzen's theory on planned behavior (i.e. social norms) [33] and Bandura's Social Cognitive Theory (i.e. self-efficacy) [34] are useful frameworks to apply when designing PA behavior change interventions. However, according to Prochaska and DiClemente's Transtheoretical Model [35], one would expect that active participation to the dance sessions would lead to a progression in stage of readiness for change towards PA, but our data did not show evidence to support it.

**Table 5. The effect of Bollywood Dance Fitness on physical activity behavioral social cognitive determinants.**

| Construct / Item | Measurement | | | | | | Diff pre/post - P-Value |
|---|---|---|---|---|---|---|---|
| **ATTITUDE** | | | | | | | |
| *> 30 min/day PA\* is* | | *Unimportant* | *Bit unimportant* | *Neutral* | *Bit important* | *Important* | 0.30 |
| | Pre: % (n) | 0 (0) | 0 (0) | 0 (0) | 19 (5) | 81 (21) | |
| | Post: % (n) | 0 (0) | 0 (0) | 0 (0) | 8 (2) | 92 (24) | |
| *> 30 min/day PA\* is* | | *Unpleasant* | *Bit unpleasant* | *Neutral* | *Bit pleasant* | *Pleasant* | 0.96 |
| | Pre: % (n) | 0 (0) | 4 (1) | 4 (1) | 19 (5) | 73 (19) | |
| | Post: % (n) | 0 (0) | 0 (0) | 4 (1) | 31 (8) | 65 (17) | |
| *> 30 min/day PA\* is* | | *Difficult* | *Bit difficult* | *Neutral* | *Bit easy* | *Easy* | 0.28 |
| | Pre: % (n) | 0 (0) | 12 (3) | 4 (1) | 19 (5) | 65 (17) | |
| | Post: % (n) | 0 (0) | 12 (3) | 15 (4) | 23 (6) | 50 (13) | |
| *≥ 1 day/wk sport is* | | *Unimportant* | *Bit unimportant* | *Neutral* | *Bit important* | *Important* | 0.41 |
| | Pre: % (n) | 0 (0) | 8 (2) | 0 (0) | 8 (2) | 85 (22) | |
| | Post: % (n) | 0 (0) | 0 (0) | 0 (0) | 12 (3) | 88 (23) | |
| *≥ 1 day/wk sport is* | | *Pleasant* | *Bit pleasant* | *Neutral* | *Bit unpleasant* | *Unpleasant* | 0.33 |
| | Pre: % (n) | 0 (0) | 0 (0) | 12 (3) | 15 (4) | 73 (19) | |
| | Post: % (n) | 0 (0) | 0 (0) | 0 (0) | 23 (6) | 77 (20) | |
| *Motivated for PA* | | *Unmotivated* | *Little motivated* | *Neutral* | *Bit motivated* | *Very motivated* | 0.96 |
| | Pre: % (n) | 0 (0) | 12 (3) | 4 (1) | 35 (9) | 50 (13) | |
| | Post: % (n) | 0 (0) | 8 (2) | 12 (3) | 31 (8) | 50 (13) | |
| **SELF-EFFICACY** | | | | | | | |
| Confidence for PA in general | | *No confidence* | *Little confidence* | *Neutral* | *Much confidence* | *Full confidence* | 0.52 |
| | Pre: % (n) | 0 (0) | 15 (4) | 23 (6) | 23 (6) | 38 (10) | |
| | Post: % (n) | 0 (0) | 8 (2) | 23 (6) | 31 (8) | 38 (10) | |
| PA if weather bad | | *Certainly not succeed* | *Probably not succeed* | *Neutral* | *Probably succeed* | *Certainly succeed* | 0.18 |
| | Pre: % (n) | 12 (3) | 8 (2) | 23 (6) | 15 (4) | 42 (11) | |
| | Post: % (n) | 8 (2) | 4 (1) | 15 (4) | 23 (6) | 50 (13) | |
| PA if tired | | *Certainly not succeed* | *Probably not succeed* | *Neutral* | *Probably succeed* | *Certainly succeed* | 0.07 |
| | Pre: % (n) | 12 (3) | 31 (8) | 19 (5) | 8 (2) | 31 (8) | |
| | Post: % (n) | 12 (3) | 8 (2) | 23 (6) | 23 (6) | 35 (9) | |
| PA if little time | | *Certainly not succeed* | *Probably not succeed* | *Neutral* | *Probably succeed* | *Certainly succeed* | 0.77 |
| | Pre: % (n) | 19 (5) | 19 (5) | 8 (2) | 38 (10) | 15 (4) | |
| | Post: % (n) | 15 (4) | 19 (5) | 15 (4) | 31 (8) | 19 (5) | |
| PA if no one to join | | *Certainly not succeed* | *Probably not succeed* | *Neutral* | *Probably succeed* | *Certainly succeed* | 0.07 |
| | Pre: % (n) | 12 (3) | 8 (2) | 8 (2) | 31 (8) | 42 (11) | |
| | Post: % (n) | 0 (0) | 8 (2) | 4 (1) | 31 (8) | 58 (15) | |
| PA after some weeks of no PA | | *Certainly not succeed* | *Probably not succeed* | *Neutral* | *Probably succeed* | *Certainly succeed* | 0.09 |
| | Pre: % (n) | 0 (0) | 12 (3) | 12 (3) | 38 (10) | 38 (10) | |
| | Post: % (n) | 4 (1) | 4 (1) | 4 (1) | 23 (6) | 65 (17) | |
| BWD is only for those who have good PA skills | | *Completely agree* | *Agree a bit* | *Neutral* | *Disagree a bit* | *Completely disagree* | **0.02** |
| | Pre: % (n) | 23 (6) | 15 (4) | 4 (1) | 12 (3) | 46 (12) | |
| | Post: % (n) | 12 (3) | 4 (1) | 0 (0) | 23 (6) | 62 (16) | |

*(Continued)*

**Table 5.** (Continued)

| Construct / Item | Measurement | | | | | | Diff pre/post - P-Value |
|---|---|---|---|---|---|---|---|
| **STAGE OF CHANGE** | | | | | | | |
| Enough PA now? | | No, and don't intend in 6 months | No, but intend to in 6 months | No, but intend to in 30 days | Yes active, but shorter than 6 months | Yes active, and longer than 6 months | 0.31 |
| | Pre: % (n) | 23 (6) | 19 (5) | 15 (4) | 0 (0) | 42 (11) | |
| | Post: % (n) | 15 (4) | 8 (2) | 19 (5) | 8 (2) | 50 (13) | |
| **SKILLS / CAPABILITY** | | | | | | | |
| Can make plans for PA | | Completely disagree | Disagree a bit | Neutral | Agree a bit | Completely agree | 0.43 |
| | Pre: % (n) | 8 (2) | 0 (0) | 12 (3) | 27 (7) | 54 (14) | |
| | Post: % (n) | 8 (2) | 15 (4) | 0 (0) | 27 (7) | 50 (13) | |
| Can set goals for PA | | Completely disagree | Disagree a bit | Neutral | Agree a bit | Completely agree | 0.32 |
| | Pre: % (n) | 4 (1) | 15 (4) | 8 (2) | 31 (8) | 42 (11) | |
| | Post: % (n) | 8 (2) | 4 (1) | 4 (1) | 31 (8) | 54 (14) | |
| Can make agreement with myself for PA | | Completely disagree | Disagree a bit | Neutral | Agree a bit | Completely agree | 0.25 |
| | Pre: % (n) | 4 (1) | 12 (3) | 8 (2) | 27 (7) | 50 (13) | |
| | Post: % (n) | 4 (1) | 4 (1) | 8 (2) | 23 (6) | 62 (16) | |
| Can join BWD lessons 2x a week | | Completely disagree | Disagree a bit | Neutral | Agree a bit | Completely agree | 0.09 |
| | Pre: % (n) | 0 (0) | 0 (0) | 4 (1) | 15 (4) | 84 (21) | |
| | Post: % (n) | 0 (0) | 0 (0) | 0 (0) | 4 (1) | 96 (25) | |
| Can participate in dance sessions, even when tempo high | | Completely disagree | Disagree a bit | Neutral | Agree a bit | Completely agree | 0.36 |
| | Pre: % (n) | 0 (0) | 4 (1) | 4 (1) | 46 (12) | 46 (12) | |
| | Post: % (n) | 0 (0) | 8 (2) | 4 (1) | 19 (5) | 69 (18) | |
| Can perform all dance moves | | Completely disagree | Disagree a bit | Neutral | Agree a bit | Completely agree | 0.13 |
| | Pre: % (n) | 12 (3) | 15 (4) | 27 (7) | 38 (10) | 8 (2) | |
| | Post: % (n) | 8 (2) | 15 (4) | 12 (3) | 50 (13) | 15 (4) | |
| Have enough stamina for BWD | | Completely disagree | Disagree a bit | Neutral | Agree a bit | Completely agree | **0.01** |
| | Pre: % (n) | 0 (0) | 15 (4) | 8 (2) | 12 (3) | 65 (17) | |
| | Post: % (n) | 0 (0) | 0 (0) | 0 (0) | 8 (2) | 92 (24) | |
| **PERCEIVED SOCIAL NORM (Important others . . .)** | | | | | | | |
| . . . think that I should have regular PA | | Completely disagree | Disagree a bit | Neutral | Agree a bit | Completely agree | **0.01** |
| | Pre: % (n) | 8 (2) | 24 (6) | 16 (4) | 20 (5) | 32 (8) | |
| | Post: % (n) | 4 (1) | 4 (1) | 12 (3) | 31 (8) | 50 (13) | |
| . . . support me to have regular PA | | Completely disagree | Disagree a bit | Neutral | Agree a bit | Completely agree | 0.09 |
| | Pre: % (n) | 4 (1) | 31 (8) | 12 (3) | 12 (3) | 42 (11) | |
| | Post: % (n) | 8 (2) | 8 (2) | 12 (3) | 19 (5) | 54 (14) | |
| . . . have regular PA themselves | | Completely disagree | Disagree a bit | Neutral | Agree a bit | Completely agree | 0.08 |
| | Pre: % (n) | 24 (6) | 15 (4) | 8 (2) | 8 (2) | 46 (12) | |
| | Post: % (n) | 8 (2) | 15 (4) | 4 (1) | 15 (4) | 58 (15) | |
| SOCIAL SUPPORT . . . | | | | | | | |
| . . .from partner[1] | | No, never | No, almost never | Yes, sometimes | Yes, often | Yes, very often | 0.77 |

(Continued)

**Table 5.** (Continued)

| Construct / Item | Measurement | | | | | | Diff pre/post - P-Value |
|---|---|---|---|---|---|---|---|
| | Pre: % (n) | 8 (2) | 4 (1) | 19 (5) | 15 (4) | 8 (2) | |
| | Post: % (n) | 12 (3) | 4 (1) | 8 (2) | 12 (3) | 12 (3) | |
| . . .from other family members | | *No, never* | *No, almost never* | *Yes, sometimes* | *Yes, often* | *Yes, very often* | 0.84 |
| | Pre: % (n) | 15 (4) | 8 (2) | 27 (7) | 31 (8) | 19 (5) | |
| | Post: % (n) | 15 (4) | 12 (3) | 19 (5) | 27 (7) | 27 (7) | |
| . . .from important others | | *No, never* | *No, almost never* | *Yes, sometimes* | *Yes, often* | *Yes, very often* | 0.28 |
| | Pre: % (n) | 19 (5) | 0 (0) | 42 (11) | 15 (4) | 23 (6) | |
| | Post: % (n) | 8 (2) | 12 (3) | 31 (8) | 31 (8) | 19 (5) | |
| . . .from other BWD participants[2] | | *No, never* | *No, almost never* | *Yes, sometimes* | *Yes, often* | *Yes, very often* | 0.07 |
| | Pre: % (n) | 12 (3) | 4 (1) | 19 (5) | 27 (7) | 15 (4) | |
| | Post: % (n) | 19 (5) | 0 (0) | 19 (5) | 35 (9) | 23 (6) | |
| **PERCEIVED HEALTH** | | | | | | | |
| *How healthy do you feel?* | | *Very poor* | *Poor* | *Neutral* | *Good* | *Very good* | 0.48 |
| | Pre: % (n) | 0 (0) | 0 (0) | 42 (11) | 46 (12) | 12 (3) | |
| | Post: % (n) | 0 (0) | 0 (0) | 31 (8) | 62 (16) | 8 (2) | |

Note: Bold numbers represent a statistically significant effect of p<0.05. Abbreviations: BWD: Bollywood Dance Fitness; PA: Physical Activity. PA is defined as ≥ 30 min/day, 5 days a week leisure time moderate physical activity and/or sports; PA* is defined as all activities during daily life, but sports excluded.

[1]Not applicable at pretest 46%(n = 12) and posttest: 54%(n = 14);

[2]Not applicable at pretest 23%(n = 6) and posttest 4%(n = 1).

The Bollywood Dance Fitness sessions showed a substantial variation in MVPA between participants during a dance session, suggesting that participation by itself does not necessarily increase MVPA for all participants. However, since motor skills may have increased and instruction time decreased during the sessions we already observed on average a 28% increase of MVPA intensity level after 19 sessions, whereby those that attended more sessions showed a higher increase in MVPA. Twice a week Bollywood Dance Fitness can help women of SA origin to obtain the recommended amount of 150 minutes a week of MVPA as is suggested in Dutch guidelines (i.e. four additional women fulfilled post program this recommended level) [12].

Half of the increase in MVPA was explained due to the Bollywood Dance Fitness, although there was a non-significant tendency that participants also spent more time of their daily life in MVPA (i.e. if we did not take the Bollywood Dance Fitness session data into account, a non-significant increase of 3 minutes of MVPA is left). During focus groups participants indicated that they were challenged to dance at home or during celebrations with friends.

This is the first study in Europe on implementation of Bollywood Dance for health promotion among women of SA origin. This study was able to reach participants from all social economic classes which is something other lifestyle intervention studies struggle with. Another strength is the objective assessment of PA during the dance sessions, before the start of the intervention, and at the end of the intervention, which resulted in more accurate assessment compared to self-report measures. It is important to consider that our study was designed as a feasibility study with a relatively small sample size and consequently limited power to detect significant differences or to conduct subgroup analyses, for example focusing on the small

group of participants who did not like the intervention or did not feel that the intervention enhanced their capability to be physically active. This would be an interesting group to study more closely in any further intervention or in a future trial. Furthermore, the lack of a control group limited conclusions on intervention-effectiveness. Generalizability is limited due to a) the non-randomized sampling, which may have introduced some selection bias, and b) due to inclusion of SA women from Surinamese origin and mostly a Hindu religion, under-representing women of SA origin with other ethnic background and religions. Social desirability bias might have been introduced due to the self-reported nature of questions on behavioral determinants. And finally, accelerometers are attached to the hip and do not register upper bodily movement correctly, it is therefore possible that PA intensity-levels during the dance sessions may have been somewhat underestimated, especially the strength exercises.

## Conclusions

This study underscores the importance of community engagement in developing, planning and implementation of culturally acceptable PA promotion in enabling individual behavior change. The evidence from this study supports the idea that in line with small-scale trials in North America [17, 18], Bollywood dancing appears to be a feasible, culturally acceptable and potentially effective approach to increase PA in SA women in the Netherlands. Bollywood dancing may be an option to be used as PA component in the implementation of T2D prevention programs for communities of SA origin in the Netherlands and perhaps elsewhere in Europe. This study also demonstrates that the energy expenditure in Bollywood Dance Fitness classes exposes people to light-to-moderate intensity levels and that there is room for increasing to higher levels of intensity, when participants' skills improve and can match the intensity achieved by the instructor. A pilot cluster RCT is needed to confirm these initial findings [36].

## Supporting information

**S1 Table. Topic-guides for interviews with participants, instructor, community leader.**
(DOCX)

**S2 Table. COREQ (COnsolidated criteria for REporting Qualitative research) checklist.**
(DOCX)

## Acknowledgments

The authors would like to thank Dachel Dominique (DD) and André Bhola (AB) who took part in the organization of the Bollywood Dance Fitness program, and all participants for taking part in this study.

## Author Contributions

**Conceptualization:** Erik Beune, Irene van Valkengoed, Bernadette Kumar, Esperanza Diaz, Jason M. R. Gill, Anne Karen Jenum, Aziz Sheikh, Emma Davidson, Karien Stronks.

**Formal analysis:** Erik Beune, Mirthe Muilwijk, Judith G. M. Jelsma.

**Investigation:** Erik Beune, Annemarie M. Teitsma-Jansen.

**Methodology:** Erik Beune, Bernadette Kumar, Esperanza Diaz, Jason M. R. Gill, Anne Karen Jenum, Latha Palaniappan, Hidde P. van der Ploeg, Aziz Sheikh, Emma Davidson, Karien Stronks.

**Project administration:** Erik Beune, Annemarie M. Teitsma-Jansen.

**Writing – original draft:** Erik Beune.

**Writing – review & editing:** Mirthe Muilwijk, Judith G. M. Jelsma, Irene van Valkengoed, Bernadette Kumar, Esperanza Diaz, Jason M. R. Gill, Anne Karen Jenum, Latha Palaniappan, Hidde P. van der Ploeg, Aziz Sheikh, Emma Davidson, Karien Stronks.

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
