## [Decision Letter · Decision Letter 0]

7 Sep 2021

PONE-D-21-14454The acceptability and effect of a culturally-tailored dance intervention to promote physical activity to prevent type 2 diabetes in women of South Asian origin in the Netherlands – a mixed-methods feasibility studyPLOS ONE

Dear Dr. Beune,

Thank you for submitting your manuscript to PLOS ONE. After careful consideration, we feel that it has merit but does not fully meet PLOS ONE’s publication criteria as it currently stands. Therefore, we invite you to submit a revised version of the manuscript that addresses the points raised during the review process.

Although I have marked it as major revision, I think you will see from the reviewers comments that most of the revisions requested are manageable.  I would like you to strengthen the discussion as requested by one of the reviewers.

We look forward to receiving your revised manuscript.

Kind regards,

Ruth Jepson, PhD

Academic Editor

PLOS ONE

Journal Requirements:

Reviewers' comments:

Reviewer's Responses to Questions

**Comments to the Author**

1. Is the manuscript technically sound, and do the data support the conclusions?

Reviewer #1: Yes

Reviewer #2: Yes

2. Has the statistical analysis been performed appropriately and rigorously? 

Reviewer #1: Yes

Reviewer #2: Yes

3. Have the authors made all data underlying the findings in their manuscript fully available?

Reviewer #1: Yes

Reviewer #2: No

4. Is the manuscript presented in an intelligible fashion and written in standard English?

Reviewer #1: Yes

Reviewer #2: Yes

5. Review Comments to the Author

Reviewer #1: This study is important. Identifying sub-groups and ways to enhance physical activity will be steadily more important. I especially appreciate the mixed-method approach, as it gives a broader range of data to use. The quantitative data are well analysed, and the use of GEE is a strength.

My main – and – only criticism is that the Discussion is not up to the standards. I would suggest that You interpret Your finding in the light of theories of health behaviour, and since You already have data on Social norms and skills, Ajzen’s Theory of Planned behaviour may be beneficial. Also, since You have data on readiness for change, Prochaska and DiClemete’s Transtheorietic Model may be handy to use. I miss more analysis of those in the low end of the scales, those who did not like or felt capable. That is a small, but an interesting group. What happened to them? Are they still “PA resistant”?

I suggest You remove “prevent type 2 diabetes” from the title, as PA can be used for more than that, and You cannot “tailor” Your intervention to only influence DT2 risks. What You say is Your aim is more to the point.

Table 4: the two last answer alternatives to “Enough PA?” is unclear to me (Yes, shorter than 6 months…), perhaps rephrase to: Yes, active, but shorter than… ?

Some more suggestions in the attached file

Reviewer #2: This is an important study that develops a culturally tailored dance intervention, and evaluates acceptability and effect of the intervention. The study also throws light on the intensity of the dance sessions, which is important to understand how the intervention can promote health outcomes, as well as adherence to physical activity recommendations. Some suggestions and recommendations below:

1. Line 67: However, our analysis of available interventions for T2D prevention in populations of SA origin in Europe has found that in particular the PA element has been underused (e.g. high drop-out rates) [9]

- It is not immediately clear if the physical activity component was underused within interventions (ie. not many interventions have a physical activity component), or the uptake/adoption was low among participants. It would be good to clarify this. It would also be good to have more evidence to support this if possible- or is it that there are few interventions to begin with?

2. Can the introduction section address and introduce some of the outcome measures such as PA guidelines (would be a good way to introduce intensities), as well as social cognitive determinants.

3. Line 139: Implementation was supported by a written protocol for the dance instructor combined with two 120 minutes face-to-face training sessions, with the second one including feedback, based on structured observations from some of the dance classes.

- Can you please elaborate on who conducted the training? It would also be good to get more details on the feedback based on structured observations- who provided the feedback and on what (was it on the teaching method, or class structure); were any tools used for the structured feedback?

4. Methods, data collection, qualitative data

- Can more detail be provided on how the topic guides for interviews and focus groups were developed, and please provide the topic guides as supplementary material (if not already provided).

5. Methods, data collection- qualitative data

- Can you please provide more details on who conducted the interviews and focus groups? Were there moderators and co-moderators?

6. Methods, data collection- qualitative data

- Please use the COREQ checklist to ensure that reporting of methods/results for qualitative data is rigorous.

7. Methods, data collection- quantitative data

- How were the questionnaire administered- online/paper? Any software used?

8. In the qualitative analysis and results, I would suggest removing wordings such as “It also emerged that…” (line 308), or “the qualitative data revealed..” (line 291). This suggests that themes passively emerge, and reduce the researcher role in analysing, creating and interpreting themes (Virginia Braun & Victoria Clarke (2019) Reflecting on reflexive thematic analysis, Qualitative Research in Sport, Exercise and Health)

9. Results, qualitative data

- Can the quotes have pseudonym and age along with the focus group- so that we know the quotes are from a diverse group, and not the same person.

10. Results- PA intensity during the program (lines 424 to 432)

- It is not clear why the instructor’s physical activity collected over sessions 17, 18, 19 (light, moderate, vigorous) and comparing this with participants physical activity is important or relevant. I may be missing something- I would suggest justifying this, and if it does not add anything, please remove from methods and results (please note, the participants’ physical activity and intensities during the dance session is valuable information and should be retained. I also think highlighting variations among participants is important. However, the instructor’s data over the last 3 days may be an outlier and not representative of participants).

6. PLOS authors have the option to publish the peer review history of their article (what does this mean?). If published, this will include your full peer review and any attached files.

Reviewer #1: **Yes: **Asgeir Mamen

Reviewer #2: No

---

## [Author Response · Author response to Decision Letter 0]

2 Nov 2021

Response to the reviewers and changes made in the manuscript, based on reviewers comments

- Page and lines mentioned here refer to the file: “Revised Manuscript with Track Changes”

Reviewer #1: 

Reviewer: This study is important. Identifying sub-groups and ways to enhance physical activity will be steadily more important. I especially appreciate the mixed-method approach, as it gives a broader range of data to use. The quantitative data are well analysed, and the use of GEE is a strength.

Response: We are happy to hear and thank the reviewer for the positive feedback, including underscoring that our study is important and, in particular, appreciation for the mixed methods approach

Reviewer: My main – and – only criticism is that the Discussion is not up to the standards. I would suggest that You interpret Your finding in the light of theories of health behaviour, and since You already have data on Social norms and skills, Ajzen’s Theory of Planned behaviour may be beneficial. Also , since You have data on readiness for change, Prochaska and DiClemente’s Transtheorietic Model may be handy to use. 

Response: Thank you for your constructive comments. We agree that the discussion could address this more fully. In the method section, we explained the theoretical elements (Socio Cognitive Theory, Bandura) for the intervention (described in line 157-162) and in the discussion section, we compared our findings to other studies. However, we have now addressed other theories of behavior change more explicitly in the discussion section (page 23-24, lines 502-512):

“As outlined in the methods, our study was based on behavioral social cognitive determinants identified by evidence synthesis, and Delphi expert consensus, to be important for the initiation and maintenance of PA [14, 23]. We measured changes in these PA-behavioral social cognitive determinants and our findings support some specific models of behavior change but are not consistent with others. For example, our results are in line with and underscore findings from the literature demonstrating that Ajzen’s theory on planned behavior (i.e. social norms) [33] and Bandura’s Social Cognitive Theory (i.e. self-efficacy) [34] are useful frameworks to apply when designing PA behavior change interventions. However, according to Prochaska and DiClemente’s Transtheoretical Model [35], one would expect that active participation to the dance sessions would lead to a progression in stage of readiness for change towards PA, but our data did not show evidence to support it.”

Reviewer: I miss more analysis of those in the low end of the scales, those who did not like or felt capable. That is a small, but an interesting group. What happened to them? Are they still “PA resistant”?

Response: We agree with the reviewer that this is an interesting research group. The group is too small, however, and the information available too limited to draw any conclusion on this group. We therefore addressed this comment in the limitation section (page 25, lines 535-538):

.. limited power to detect significant differences “or to conduct subgroup analyses, for example focusing on the small group of participants who did not like the intervention or did not feel that the intervention enhanced their capability to be physically active. This would be an interesting group to study more closely in any further intervention or in a future trial.” 

Reviewer: I suggest You remove “prevent type 2 diabetes” from the title, as PA can be used for more than that, and You cannot “tailor” Your intervention to only influence DT2 risks. What You say is Your aim is more to the point.

Response: We thank the reviewer for this comment. We believe the setting of the Bollywood Dance Fitness study within the larger project (EuroDHYAN) means that our study is in the context of diabetes prevention. However, we also see -and we agree with the reviewer- that the PA changes could also prevent other conditions. We therefore addressed this comment in the title by changing the wording to reflect this and by taking out the word prevent:

Title: “The acceptability and effect of a culturally-tailored dance intervention to promote physical activity in women of South Asian origin at risk of diabetes living in the Netherlands – a mixed-methods feasibility study” 

Short title: “Mixed-methods feasibility study of culturally tailored dance intervention to promote physical activity”

Reviewer: Table 4: the two last answer alternatives to “Enough PA?” is unclear to me (Yes, shorter than 6 months…), perhaps rephrase to: Yes, active, but shorter than… ?

Response: We have changed this in the manuscript in Table 4 as suggested by the reviewer.

Reviewer: Some more suggestions in the attached file

Response: We thank the reviewer for the suggested corrections. The corrections have been made throughout the manuscript accordingly. There were however two that we weren’t sure what the highlighting referred to – in the introduction the highlight on ‘aim’ and in the conclusion there are two references highlighted and a comment but nothing written.

We thank the reviewer for the useful suggestions. These have been addressed by adding the requested information to the manuscript:

Suggestion: Software used?

Response: Page 10, lines 182-183: “All interviews were audio-recorded and transcribed verbatim without the use of software.”

Suggestion: This choice should be explained

Response: Page 11, lines 213-214: “The choice of 60-second epochs was based on the advice to use the epoch cut off with which the cut offs were designed [25].”

Suggestion: Here a reference for this claim is needed

Response: Page 11, lines 219-220: “Participants who had at least four days of valid data were included in the analyses [25, 27].” (25. Migueles JH, Cadenas-Sanchez C, Ekelund U, Nyström CD, Mora-Gonzalez J, Löf M, et al. Accelerometer data collection and processing criteria to assess physical activity and other outcomes: a systematic review and practical considerations. Sports medicine. 2017;47(9):1821-45. And 27. Trost SG, Mciver KL, Pate RR. Conducting accelerometer-based activity assessments in field-based research. Medicine & Science in Sports & Exercise. 2005;37(11):S531-S43.)

#Reviewer 2

Reviewer: This is an important study that develops a culturally tailored dance intervention, and evaluates acceptability and effect of the intervention. The study also throws light on the intensity of the dance sessions, which is important to understand how the intervention can promote health outcomes, as well as adherence to physical activity recommendations. Some suggestions and recommendations below:

Response: We thank the reviewer for this positive feedback, and for highlighting how our study, in examining PA intensity, helps to understand the potential health outcomes of the intervention, as well as adherence to physical activity recommendations.

Reviewer: comment 1. Line 67: However, our analysis of available interventions for T2D prevention in populations of SA origin in Europe has found that in particular the PA element has been underused (e.g. high drop-out rates) [9]

- It is not immediately clear if the physical activity component was underused within interventions (ie. not many interventions have a physical activity component), or the uptake/adoption was low among participants. It would be good to clarify this. It would also be good to have more evidence to support this if possible- or is it that there are few interventions to begin with?

Response: We thank the reviewer for this comment and agree with the reviewer that it was not clear. Therefore, we have clarified that the experience with these type of interventions is limited, as supported by the literature (page 5, lines 69-72).

… populations of SA origin in Europe has found that “there is limited experience of PA interventions specifically adapted for people of South Asian origin [9, 10], and within the studies in South Asians including a PA component, the PA element is often underused by participants (e.g. high drop-out rates) [9, 11].”

Reviewer: comment 2. Can the introduction section address and introduce some of the outcome measures such as PA guidelines (would be a good way to introduce intensities), as well as social cognitive determinants.

Response: Thank you for these constructive comments. We agree that the introduction section can be clearer on this. We have included a separate paragraph on this on page 5, lines 72-75:

“WHO guidelines on physical activity and sedentary behavior recommend that all adults undertake 150-300 minutes of moderate-intensity or 75-150 minutes of vigorous-intensity aerobic physical activity per week or an equivalent combination of these [12].”

and on page 6, lines 81-83:

“Psychological, social and cultural factors (e.g. self-efficacy, social support, perceived social norms) play an important role in PA initiation and maintenance and should be taken into account when designing PA behavior change interventions [14].”

Reviewer: comment 3. Line 139: Implementation was supported by a written protocol for the dance instructor combined with two 120 minutes face-to-face training sessions, with the second one including feedback, based on structured observations from some of the dance classes. - Can you please elaborate on who conducted the training? It would also be good to get more details on the feedback based on structured observations- who provided the feedback and on what (was it on the teaching method, or class structure); were any tools used for the structured feedback?

Response: We thank the reviewer for the useful suggestions. We have added the requested information in the text (page 8-9, lines 149-152): 

… with two 120 minutes face-to-face training sessions “provided by EB , with the second one including feedback, based on structured observations from some of the dance classes, including context (barriers, facilitators), participants (performance, intensity, pleasure, confidence, social interaction) and teaching (attitudes, performance, learning strategies, adherence to protocol).”

Reviewer: comment 4. Methods, data collection, qualitative data

- Can more detail be provided on how the topic guides for interviews and focus groups were developed, and please provide the topic guides as supplementary material (if not already provided).

Response: The topic guides were derived from research question 1 and 2, exploring issues related to acceptability, feasibility and experiences with the Bollywood Dance Fitness program. As suggested by the reviewer, we have provided the topic guides as supplementary material (page 38).

Reviewer: comment 5. Methods, data collection- qualitative data

 - Can you please provide more details on who conducted the interviews and focus groups? Were there moderators and co-moderators?

We have changed this in the manuscript as suggested by the reviewer (page 9-10, lines 175-180):

“Two post-intervention focus group discussions (FGD) (n=10, n=8, respectively) conducted by EB (moderator) and AT (co-moderator) with participants, two interviews (mid-term and post-intervention) conducted by EB with the Bollywood Dance Fitness instructor (DD), and one post-intervention interview conducted by EB with the community leader (AB), were used for data collection on acceptability, feasibility and experiences with the program. All interviews were held at the community center, in Dutch.”

Reviewer: comment 6. Methods, data collection- qualitative data

- Please use the COREQ checklist to ensure that reporting of methods/results for qualitative data is rigorous COREQ checklist 

Response: We have included the COREQ checklist as a separate file as part of the submission. Please note that the qualitative data collection was part of a pilot mixed method study, and that the qualitative data were used to inform observed findings in quantitative data and not as such collected and analyzed as an independent qualitative study.

Reviewer: comment 7. Methods, data collection- quantitative data

 - How were the questionnaire administered- online/paper? Any software used?

Response: We have changed this in the manuscript as suggested by the reviewer (page 10, line 187): 

“Face-to-face structured questionnaires were collected at baseline, administered on paper”

Reviewer: comment 8. In the qualitative analysis and results, I would suggest removing wordings such as “It also emerged that…” (line 308), or “the qualitative data revealed..” (line 291). This suggests that themes passively emerge, and reduce the researcher role in analysing, creating and interpreting themes (Virginia Braun & Victoria Clarke (2019) Reflecting on reflexive thematic analysis, Qualitative Research in Sport, Exercise and Health)

Response: We have changed this throughout the manuscript as suggested by the reviewer.

Reviewer: comment 9. Results, qualitative data

- Can the quotes have pseudonym and age along with the focus group- so that we know the quotes are from a diverse group, and not the same person.

Response: We have changed this throughout the manuscript as suggested by the reviewer.

We trust that these revisions are to your satisfaction, but please don’t hesitate to contact us if you require any further information. We look forward to your decision in due course. 

Sincerely,

Erik Beune, on behalf of the co-authors

---

## [Decision Letter · Decision Letter 1]

30 Dec 2021

PONE-D-21-14454R1The acceptability and effect of a culturally-tailored dance intervention to promote physical activity in women of South Asian origin at risk of diabetes in the Netherlands – a mixed-methods feasibility studyPLOS ONE

Dear Dr. Beune,

Thank you for submitting your manuscript to PLOS ONE. After careful consideration, we feel that it has merit but does not fully meet PLOS ONE’s publication criteria as it currently stands. Therefore, we invite you to submit a revised version of the manuscript that addresses the points raised during the review process.

This revised manuscript has been assessed by the two reviewers, and their comments are available below. The reviewers are positive about the work, and we request that you please address the remaining concern raised by Reviewer 2.

We look forward to receiving your revised manuscript.

Kind regards,

Vanessa Carels

Staff Editor

PLOS ONE

Journal Requirements:

Reviewers' comments:

Reviewer's Responses to Questions

**Comments to the Author**

1. If the authors have adequately addressed your comments raised in a previous round of review and you feel that this manuscript is now acceptable for publication, you may indicate that here to bypass the “Comments to the Author” section, enter your conflict of interest statement in the “Confidential to Editor” section, and submit your "Accept" recommendation.

Reviewer #1: All comments have been addressed

Reviewer #2: (No Response)

2. Is the manuscript technically sound, and do the data support the conclusions?

Reviewer #1: Yes

Reviewer #2: Yes

3. Has the statistical analysis been performed appropriately and rigorously? 

Reviewer #1: Yes

Reviewer #2: Yes

4. Have the authors made all data underlying the findings in their manuscript fully available?

Reviewer #1: (No Response)

Reviewer #2: Yes

5. Is the manuscript presented in an intelligible fashion and written in standard English?

Reviewer #1: (No Response)

Reviewer #2: Yes

6. Review Comments to the Author

Reviewer #1: Thank You for a thorough revision! I have mo more comments, except that I guess You used some kind of software in the trancription process...

Reviewer #2: Thank you for addressing the comments. The additional details add clarity and rigour to the manuscript. However, I don't think the final comment has been addressed, and I have added this again:

10. Results- PA intensity during the program (lines 424 to 432) - It is not clear why the instructor’s physical activity collected over sessions 17, 18, 19 (light, moderate, vigorous) and comparing this with participants physical activity is important or relevant. I may be missing something- I would suggest justifying this, and if it does not add anything, please remove from methods and results (please note, the participants’ physical activity and intensities during the dance session is valuable information and should be retained. I also think highlighting variations among participants is important. However, the instructor’s data over the last 3 days may be an outlier and not representative of participants).

7. PLOS authors have the option to publish the peer review history of their article (what does this mean?). If published, this will include your full peer review and any attached files.

Reviewer #1: **Yes: **Asgeir Mamen

Reviewer #2: No

---

## [Author Response · Author response to Decision Letter 1]

7 Jan 2022

Response to the reviewers and changes made in the manuscript, based on reviewers comments

- Page and lines mentioned here refer to the file: “Revised Manuscript with Track Changes”

Reviewer #1: Thank You for a thorough revision! I have mo more comments, except that I guess You used some kind of software in the trancription process.

Response: We are happy to hear and thank the reviewer for the positive feedback. 

We have clarified that we did not use additional audio transcription software in the Materials and methods section (page 10, line 181): “All interviews were audio-recorded and transcribed verbatim without the use of audio transcription software.”

Reviewer #2: Reviewer #2: Thank you for addressing the comments. The additional details add clarity and rigour to the manuscript. However, I don't think the final comment has been addressed, and I have added this again: 

10. Results- PA intensity during the program (lines 424 to 432) - It is not clear why the instructor’s physical activity collected over sessions 17, 18, 19 (light, moderate, vigorous) and comparing this with participants physical activity is important or relevant. I may be missing something- I would suggest justifying this, and if it does not add anything, please remove from methods and results (please note, the participants’ physical activity and intensities during the dance session is valuable information and should be retained. I also think highlighting variations among participants is important. However, the instructor’s data over the last 3 days may be an outlier and not representative of participants).

Response: We thank the reviewer for the positive feedback, including underscoring that the additional details add clarity and rigour to the manuscript. It was an oversight that the final comment has not been addressed previously and we apologize for that. As suggested by the reviewer we agree that it may add clarity to justify why we collected the instructor’s physical activity data. 

We addressed this comment in the Materials and methods section (page 11, lines 219-221): “This made it possible to identify the potential intensity-level if participants would achieve ideal performance (like the instructor).” In addition, we specified in the Conclusion section (page 25, line 552): … when participants’ skills improve “and can match the intensity achieved by the instructor.”

---

## [Decision Letter · Decision Letter 2]

7 Feb 2022

The acceptability and effect of a culturally-tailored dance intervention to promote physical activity in women of South Asian origin at risk of diabetes in the Netherlands – a mixed-methods feasibility study

PONE-D-21-14454R2

Dear Dr. Beune,

We’re pleased to inform you that your manuscript has been judged scientifically suitable for publication and will be formally accepted for publication once it meets all outstanding technical requirements.

Kind regards,

Vanessa Carels

Staff Editor

PLOS ONE

Additional Editor Comments (optional):

Reviewers' comments:

Reviewer's Responses to Questions

**Comments to the Author**

1. If the authors have adequately addressed your comments raised in a previous round of review and you feel that this manuscript is now acceptable for publication, you may indicate that here to bypass the “Comments to the Author” section, enter your conflict of interest statement in the “Confidential to Editor” section, and submit your "Accept" recommendation.

Reviewer #1: All comments have been addressed

Reviewer #2: All comments have been addressed

2. Is the manuscript technically sound, and do the data support the conclusions?

Reviewer #1: Yes

Reviewer #2: Yes

3. Has the statistical analysis been performed appropriately and rigorously? 

Reviewer #1: Yes

Reviewer #2: Yes

4. Have the authors made all data underlying the findings in their manuscript fully available?

Reviewer #1: Yes

Reviewer #2: Yes

5. Is the manuscript presented in an intelligible fashion and written in standard English?

Reviewer #1: Yes

Reviewer #2: Yes

6. Review Comments to the Author

Reviewer #1: This is fine! I find Your comment on why the instructor wore an accelerometer clarifying - even if this was not mine comment

Reviewer #2: Thank you for addressing all the comments. I have no further comments and happy to recommend that the manuscript be accepted for publication.

7. PLOS authors have the option to publish the peer review history of their article (what does this mean?). If published, this will include your full peer review and any attached files.

Reviewer #1: **Yes: **Asgeir Mamen

Reviewer #2: No

---

## [Editor Report · Acceptance letter]

16 Feb 2022

PONE-D-21-14454R2 

The acceptability and effect of a culturally-tailored dance intervention to promote physical activity in women of South Asian origin at risk of diabetes in the Netherlands – a mixed-methods feasibility study 

Dear Dr. Beune:

I'm pleased to inform you that your manuscript has been deemed suitable for publication in PLOS ONE. Congratulations! Your manuscript is now with our production department. 

Kind regards, 

on behalf of

Dr. Vanessa Carels 

Staff Editor

PLOS ONE